# Active colloidal molecules assembled via selective and directional bonds

Zuochen Wang [1,2], Zhisheng Wang [1,2], Jiahui Li[1], Changhao Tian[1] & Yufeng Wang [1✉]

The assembly of active and self-propelled particles is an emerging strategy to create dynamic materials otherwise impossible. However, control of the complex particle interactions remains challenging. Here, we show that various dynamic interactions of active patchy particles can be orchestrated by tuning the particle size, shape, composition, etc. This capability is manifested in establishing dynamic colloidal bonds that are highly selective and directional, which greatly expands the spectrum of colloidal structures and dynamics by assembly. For example, we demonstrate the formation of colloidal molecules with tunable bond angles and orientations. They exhibit controllable propulsion, steering, reconfiguration as well as other dynamic behaviors that collectively reflect the bond properties. The working principle is further extended to the co-assembly of synthetic particles with biological entities including living cells, giving rise to hybrid colloidal molecules of various types, for example, a colloidal carousel structure. Our strategy should enable active systems to perform sophisticated tasks in future such as selective cell treatment.

---

[1] Department of Chemistry, The University of Hong Kong, Pokfulam Road, Hong Kong SAR, China. [2] These authors contributed equally: Zuochen Wang, Zhisheng Wang. ✉email: wanglab@hku.hk

Over the last decade, the ability to control the self-assembly of colloidal particles has been significantly improved, which is largely achieved by engineering the interparticle interactions so that they are ever tunable and even programmable[1–3]. Notably, bioinspired recognition motifs such as oligonucleotides (DNA) have been introduced[4,5]. The binding strength and selectivity are determined by user-customized sequences and have led to the formation of numerous complex superlattices[6–8]. Later, the desire to assemble low-coordination, open structures for photonic applications has necessitated the realization of directional interactions. A plethora of anisotropic particles have been synthesized, including ones with different shapes (rods, triangles, cubes, etc.) as well as surface patterns (Janus and patchy particles)[9–17]. They form colloidal bonds with a high directionality, with many new structures such as lock-and-key assembly[18], Kagome lattice[19], helices[20], diamond sub-lattice[21], and colloidal molecules[9] reported. The diverse structures accomplished are expected to find applications in photonics, filtration, sensing[22–25], etc.

Recently, active colloids have become a new paradigm for colloidal science where a distinct set of particle interactions can be established. Active colloids are 'micromotors' that autonomously propel[26–30]. The propulsion is powered by external energy in various forms, such as chemical reactions, light, heat, or electric/magnetic field[31–37], which produces an imbalanced distribution of chemicals or charges around a particle. The imbalanced chemical cloud can be interactive to nearby colloids through osmotic, phoretic, dipolar, and other forces that collectively organize particles in structures of dynamic, out-of-equilibrium characteristics[38,39]. A few known examples are living crystals of photo-activated colloidal surfers[40], schools of silver chloride particles with light[41], and reconfigurable swarms of Janus particles under AC electric field[42].

The capability of self-propulsion as well as self-assembly has endowed active colloids with great promise to create dynamic materials useful for, e.g. nano-machinery and micro-machinery[43–45], where control and coordination of motions are challenging at the relevant length scales. Although much research effort has been devoted to developing particles able to propel with regulated trajectories—some bearing multiple functions[46,47], strategies for controlling the dynamic interparticle interactions for the purpose of self-assembly are less explored but emerging rapidly. For example, $TiO_2/SiO_2$ Janus micromotors have been mixed with passive spheres, which under UV irradiation induce the crystallization of passive particles by diffusiophoretic effect[48]. By using a combination of point and area illumination, hematite-polymer particles form spinning microgears, whose motions can be further synchronized[49]. Dielectric particles with a peanut shape have been assembled under AC electric field to produce chiral clusters that spins[50]. AC electric field has also been used to guide metallodielectric Janus spheres to realize multiple dynamic modes, such as swarms and propelling chains[42]. More recently, large dielectric objects with customized shapes have been shown to assemble, which in turn reveal controlled motions[51]. Despite the recent progresses, the kinds of structures obtained for active colloids are limited, far from what has been achieved for static assembly.

To expand the scope of dynamically assembled structures with active colloids, we envision the introduction of selective and directional bond, the key element that accounts for the structural versatility, fidelity, and tunability for colloidal assembly at equilibrium. Such a bonding scheme should allow active particles to assemble with improved precision and diversity, and at the same time coordinate their dynamics within the framework of the predetermined structures. While for static systems, the particle interactions and structures are programmed by the use of, e.g., DNA and surface patches[9], it remains an important issue how to control the complex interactions between active particles to facilitate the desired bonds.

Herein, we show that the dynamic interactions of active particles can be orchestrated through tuning the particle size, shape, and composition, which is then manifested in establishing colloidal bonds that are dynamic, selective, and highly directional. We demonstrate our idea in the self-assembly of colloidal molecules with the aid of AC electric field, where one or multiple metallodielectric patchy particles as ligands are bonded to a central colloid forming discrete clusters. By adjusting the size and shape of the patchy particles as well as the electric field parameter, we can readily control the properties of the colloidal bonds, their strength, selectivity, directionality (in this case, bond angle and particle orientations), formation kinetics, etc. This capability has enabled various new structures and dynamics otherwise impossible, including clusters that translate, spin, steer, and reconfigure, all reflecting the characteristic bonding attributes.

The working principle for selective and directional bonding is further extended to biological systems where synthetic particles are interfaced with bio-entities—in our case living cells—and form hybrid colloidal molecules in a prescribed manner. With these hybrid structures, dynamic modes of active particles (moving, steering, etc.) are translated to bacteria and cells, thus providing a new means for cell manipulation, transport and treatment. Moreover, because biological entities have diverse shapes (such as irregular spheres and rods) and properties (e.g., they are deformable) readily available, they serve as templates to influence the self-assembly of active particles. For example, we show the formation of a "colloidal carrousel" in which patchy particles bound to a cell constantly propel in a cyclic fashion.

## Results

**Colloidal bond selectivity.** The system we use features the assembly of colloidal clusters, or colloidal molecules, having a central dielectric sphere to which one or several ligand particles are bonded (Fig. 1a). The ligands can simply be spheres, or particles of complex surfaces and shapes. The assembly is driven by AC electric field at low frequencies, which exerts multiple interactions (e.g., dielectrophoretic, dipolar, electrohydrodynamic (EHD), electric-double layer, etc.) between the particles that collectively form the dynamic colloidal bonds. As will be discussed, the sign and/or magnitude of those field-induced interactions are sensitive to the size, composition, and shape of the particles as well as the field parameters, which serves our purpose for realizing colloidal bonds and controlling the bond selectivity and directionality.

The formation of colloidal molecules with selective bonds requires no complex particles but spheres of different sizes. When we mix dielectric spheres $D = 4\,\mu m$ (the central particle) and $d = 1\,\mu m$ (the ligand) in diameter and apply AC electric field at frequencies $f = 400–2000\,Hz$, they assemble and exclusively form heterogenous clusters, i.e., cluster consisting of both types of spheres (Fig. 1a–c, Supplementary Movie 1). When the initial feed ratio $r$ ($r =$ number ratio of 1-μm over 4-μm particles) is low, dimer clusters (or AB-type molecules) are popular. The clusters are able to propel (or swim) too, their propulsion trajectories shown in Fig. 1b. The selective bonding is further tested by mixing dielectric spheres 1, 2.5, and 4 μm in size, and the only assembly are still clusters composed of 1 and 4-μm particles (Supplementary Fig. 1 and Movie 2).

Most crucial to such size-selective bonding is dielectrophoretic interaction. Under AC electric field, dielectric particles less polarized than the medium (DI water) deflect the incident field lines[51,52]. When the particle is sited close to the electrode (substrate), a local, nonuniform electric field is generated

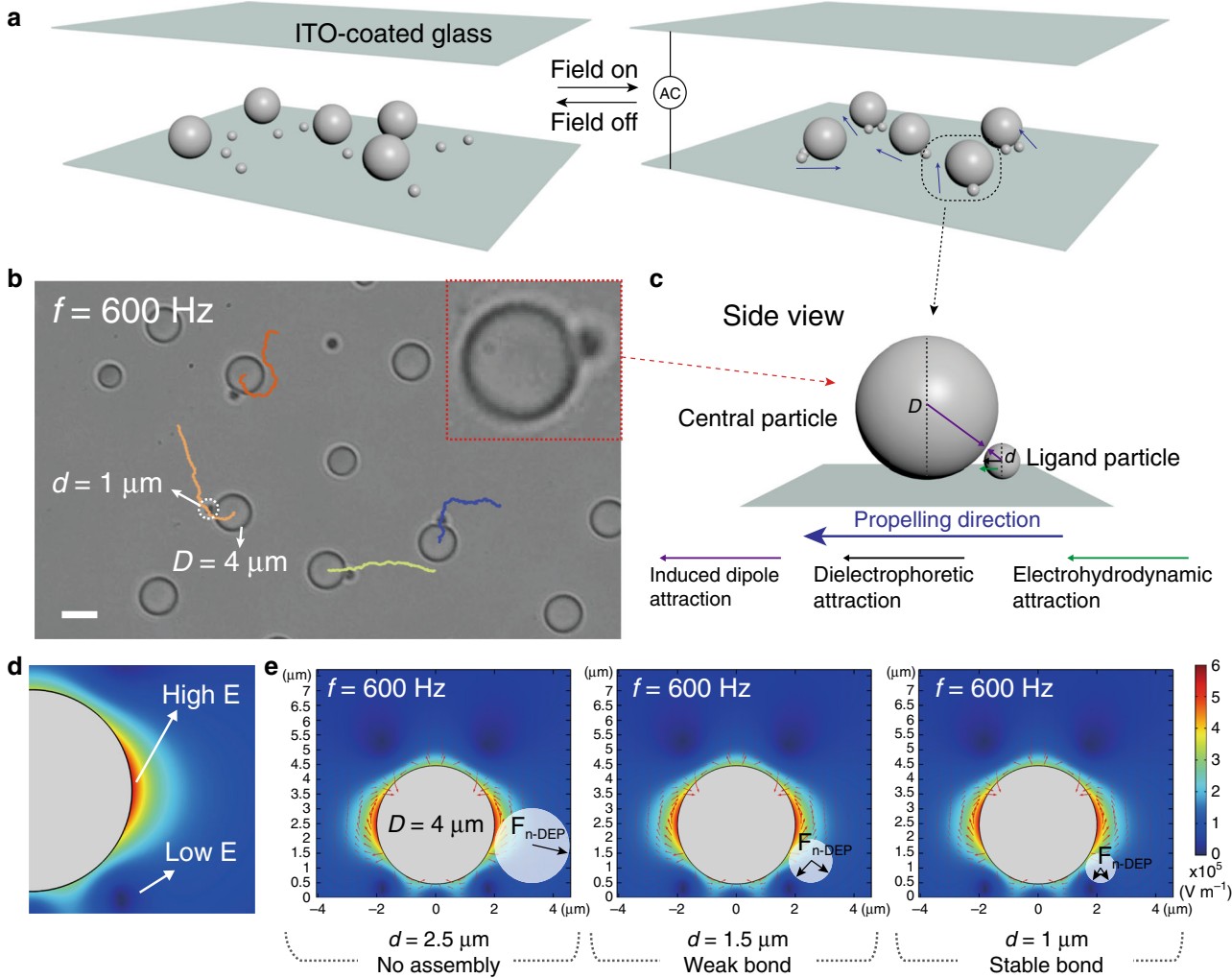

**Fig. 1 Colloidal molecules by purely dielectric spheres of different sizes. a** Schematic illustration of experimental setup for the assembly of colloidal molecules under AC electric field. **b** Optical microscope image showing the AB colloidal molecules and their propulsion trajectories. Dielectric spheres 1 and 4-μm (diameter, $d$ and $D$) are used. **c** Cartoon of an AB colloidal molecule (a side view) and various interparticle forces that account for its formation and dynamics. The larger 4-μm particle is referred to as the central sphere; the smaller 1-μm is referred to as the ligand. **d**, **e** Finite-element analysis of the nonuniform electric field locally distributed around a dielectric sphere ($D = 4\,\mu m$) at AC frequency $f = 600$ Hz, showing the high-field and low-field region ($x$–$z$ plane). A negatively polarized particle (e.g., dielectric) experiences negative dielectrophoretic (n-DEP) forces, being attracted towards the low-field region and/ or repelled by the high field according to its size. Red arrows represent the direction of electric field lines. Color bar: electric field strength. Scale bar: 4 μm.

exhibiting regions of low field, both above and beneath the particle, as well as regions of high field around the particle equator. In response to this nonuniform electric field, a second dielectric particle (the ligand) of a small size can be trapped in the low-field region, forming a bond due to negative dielectrophoretic force, or n-DEP. A bigger ligand particle can feel the high field from the equator and thus be repelled, disfavoring a bond. Instead, a metallic particle as a ligand will be attracted to the equator or avoid being attracted at the bottom part of the central particle by positive DEP force (p-DEP), according to its size.

We simulate the nonuniform electric field around the central sphere by a finite-element calculation, considering the AC response of the particle's electric double layer (EDL) (see "Methods" section). Figure 1d displays a cross section ($x$–$z$ plane) of the electric field distribution around a 4-μm polymer particle at $f = 600$ Hz, showing the referred regions of low and high field. Dielectric ligand particles of different sizes are fitted next to the central particle; the directions of DEP are shown to reveal the various bonding scenarios depending on the particle sizes (Fig. 1e).

Another factor that contributes to the size-selectivity is dipolar interaction, induced by the strong nonuniform electric field around the particle and close to the electrode at $f \leq 2000$ Hz[53]. Being bound near to the substrate, the dipolar interaction between adjacent particles is in plane, the sign and value determined by their relative sizes. That is, a small ligand particle can fit underneath the central particle and experiences a dipolar attraction; on the contrary, a larger ligand feels strong repulsion (see Supplementary Discussions 1, 2).

The nonuniform electric field also yields an EHD flow close to the conducting substrate (Fig. 2a)[54]. Under applied electric field, an excess amount of surface charges can be induced at the substrate–fluid interface. The tangential component of the nonuniform electric field around a particle acts on the induced charges and causes the EHD flow[54,55]. The direction of EHD in our case is determined to be pointing to the central sphere, which provides an additional attraction between the bound particles of a cluster (Supplementary Fig. 2, Movie 3). More importantly, the asymmetrical shape of the colloidal molecule causes an imbalanced EHD flow, which propels the clusters with the bigger

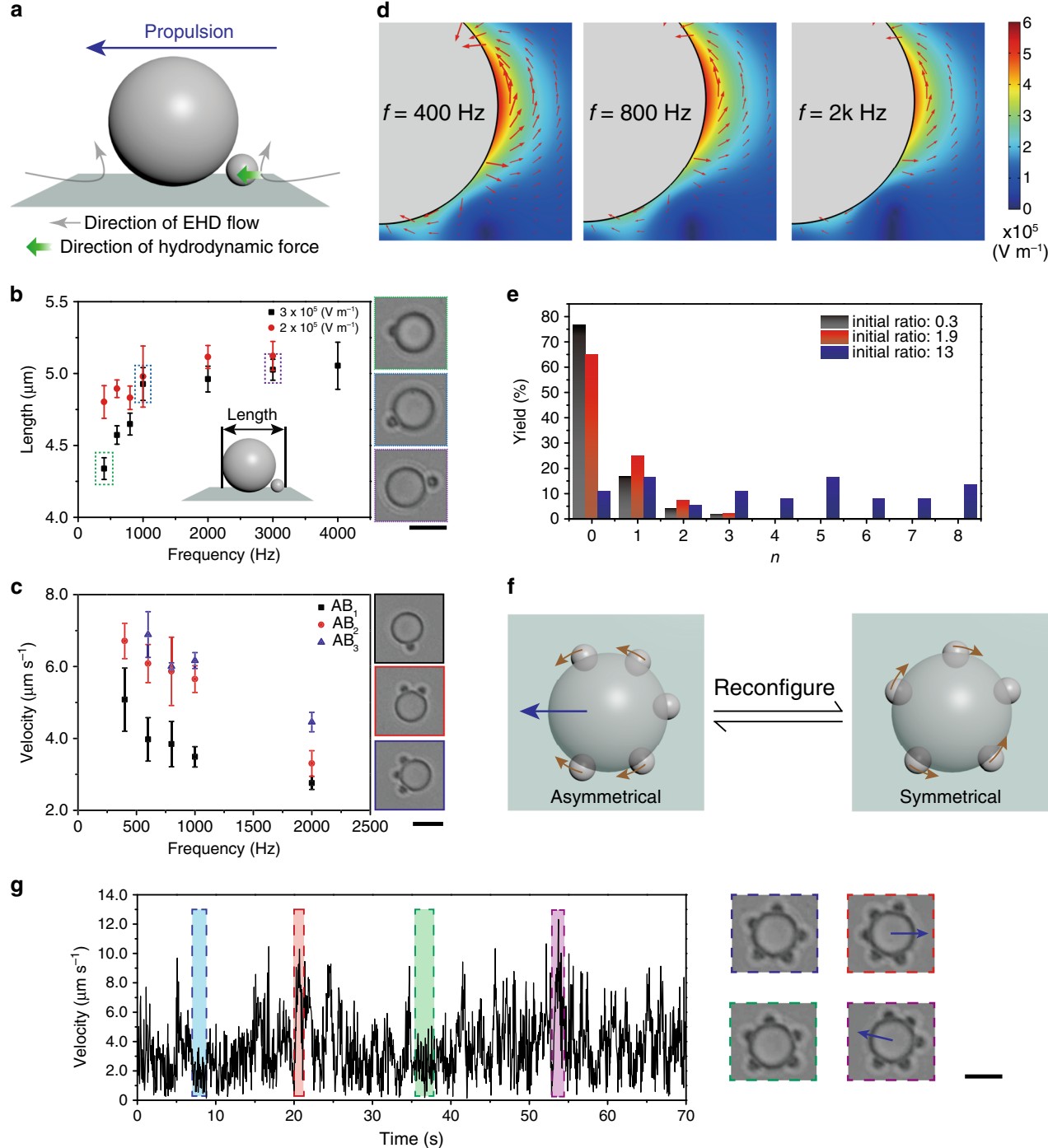

**Fig. 2 Dynamics of colloidal molecules by purely dielectric spheres. a** Schematic illustration showing the direction of electrohydrodynamic flow (EHD) around the central dielectric sphere. The EHD flow provides an additional hydrodynamic force for colloidal bonding; it also propels the colloidal molecules. **b** The bond length of colloidal molecules deceases as AC frequency decreases as labeled. **c** The propulsion velocity of colloidal molecules increases at lower AC frequencies owing to enhanced EHD flow. The velocity increases from $AB_1$ (black square) to $AB_2$ (red circle) and $AB_3$ (blue triangle) colloidal molecules due to the geometrical asymmetry. **d** Finite-element analysis showing that the nonuniformity of the electric field around a dielectric sphere becomes prominent at lower AC frequencies; DEP and EHD increase accordingly. Red arrows represent the direction of electric field lines. Color bar: electric field strength. **e** The yields of colloidal molecules of different order $n$, depending on the initial feed ratio of ligand and central colloid as labeled. **f** and **g** High-order colloidal molecules constantly reconfigure as the ligand particles slide along the rim of central sphere, which leads to the simultaneous velocity change, as highlighted and labeled by corresponding color. Scale bar: 4 μm. The error bars refer to the standard deviation of values for multiple experiments.

central particle orienting forward (Figs. 1b, 2a, Supplementary Movie 1). This is significant as micromotors are conventionally fabricated as a monolith; here, they are achieved by combining simple passive spheres, which only become active after assembly[56–58]. As EHD flow $U_{EHD}$ is nonlinearly proportional to the applied field $E$, $U_{EHD} \propto E^2$, the velocity of the propulsion follows the same trend (Supplementary Fig. 3).

We next explore the frequency ($f$) dependence of colloidal bonding and propulsion. As we decrease the AC frequency (e.g., from 2000 to 400 Hz), the bond length decreases suggesting a tighter binding situation (Fig. 2b). At the same time, the propulsion speed is increased (Fig. 2c). The observed trend can be attributed to the fact that the nonuniform electric field is more profound at low frequency at which the capacitor impedance by the EDL is more prominent, as shown by previous reports[53] and evidenced by our simulation in Fig. 2d. The enhanced nonuniformity of the induced field is accompanied by stronger DEP, dipolar, and EHD interactions (and vice versa). The imbalanced EHD flow, which propels the cluster, is inversely proportional to $f$, i.e., $U_{EHD} \propto f^{-1}$, approximately.

High-order colloidal molecules ($AB_n$, $n \geq 2$) are realized when more ligand particles are available (increased $r$). Figure 2e and Supplementary Fig. 4 survey the populations of $AB_n$ molecules as a function of $n$, at different initial ratio $r$ ($r = 0.3$, 1.9, and 13). The high-order molecules show intriguing reconfigurable shapes and dynamics. For example, all the bound ligands prefer to stay at one side and push the central particle to propel. For $AB_2$ and $AB_3$, enhanced propulsions are observed due to the increased asymmetry of the assembly (compared to AB). Their propulsion velocity also increases as AC frequency decreases (Fig. 2c). For $AB_5$ molecule, the clusters change the propulsion direction frequently as the ligand particles slide along the rim of the central particle (Fig. 2f, g, Supplementary Movie 4).

As multiple interactions are in play and delicately balanced, direct calculation of the suitable size ratios of particles for bonding is complex. The critical value is thus experimentally determined. Given a 4-μm central sphere, a ligand of 1.45 μm ($d/D = 0.36$) will be at the bonding–unbonding boundary, only forming a loose, dynamic interactions. The assembly constantly associates and falls part (Supplementary Fig. 5, Movie 5). No assembly occurs when the ligand particles are well larger than 1.5 μm. This gives us a rough reference for designing directional bonds below.

## Dielectric colloidal bond

Having shown the size-selective bonding based on simple spheres, we seek to build (selective and) directional bonds by utilizing anisotropic particles as ligands, in our case metallodielectric patchy particles each having a metallic (gold) lobe and a dielectric (polymer) lobe. The patchy particles are synthesized via a surfactant-aided dewetting method we recently developed[59]. The size of the two lobes and the particle's aspect ratio can be independently addressed (Supplementary Fig. 6, Supplementary Table 1). We note to the readers that the directionality of bonding here refers to how patchy particles approach and bind the central sphere as well as their final orientations in a colloidal molecule (different from bond directionality in real molecules). We demonstrate that both "dielectric colloidal bond" and "metallic colloidal bond" can be achieved, where either the dielectric lobe or the metallic lobe of the patchy particles is attached to the central colloid. Rich structural and dynamic behaviors are associated with the directional bonds we enable.

To form colloidal molecules with dielectric bonds, we employ patchy particles with a small polymer lobe ($d_1/D < 0.36$, $d_1$ is the size of dielectric lobe), which binds to the central sphere (size is

$D$) through n-DEP. Meanwhile, the size of the metallic lobe ($d_2$) should be properly selected so its interactions with the central particle such as p-DEP does not override the dielectric bond. Figure 3a–c show the illustrations of a typical dielectric bond; the dielectric lobe (gray color) of the patchy particle is in contact with the central sphere whereas the metallic lobe (black color) is not.

We first investigate the assembly of patchy particles $P_1$ with 4-μm central spheres to form AB colloidal molecules (Fig. 3c, d, top). $P_1$ has a polymer lobe of 1.34 μm in diameter ($d_1/D = 0.34 < 0.36$) and a metallic lobe of 1.63 μm (cartoon drawn to match SEM). Under the applied electric field, the metallic lobe is more polarized, inducing diffusive charges near its the surface. The charges responding to the AC field can generate induce-charge electro-osmosis (ICEO) flow, which propel the patchy particle under the induced-charge electrophoresis (ICEP) mechanism[60] with the dielectric lobe facing forward (Supplementary Fig. 7). At the frequency studied, ICEP is proportional to $E^2$ ($U_{ICEP} \propto E^2$) and insensitive to frequency[55]. When encountering the 4-μm sphere, $P_1$ binds to it using the dielectric lobe resulting in the desired dielectric bond. While the metallic lobe is large enough, it interacts with the central sphere through p-DEP attraction, which pulls the metallic lobe toward the central sphere but balanced by the repulsion of the induced particle dipoles and EDL. As a result, a specific bond angle can be defined, which is the angle between the tangent line of where the two particle contacts and the long axis of the patchy particle (defined in $x$ and $y$ plane). At $f = 2000$ Hz, the bond angle is 55° (Fig. 3d, e). Because the p-DEP attraction is stronger at lower frequencies (Fig. 2d), the bond angle can be adjusted by tuning the field frequency. We show that, for example, at $f = 1500$, 1000, and 800 Hz, the bond angle is 48°, 43°, and 41°, respectively (Fig. 3d, top and e). A decreased bond angle is the result of the increased p-DEP attraction, which further attracts the metallic lobe toward the central sphere and establish new balanced positions (Fig. 3b and c).

Compared to (dielectric) spheres as ligands, the use of patchy particles is in favor of the dielectric bonds. First, the metallodielectric particles are slightly more attracted to the substrate because of enhanced DEP force (from the substrate) and the asymmetrical flow surrounding the particle[60,61]. This facilitates the trapping of the dielectric lobe to the low field of the central sphere. Second, the ICEP propulsion pushes the patchy particle toward the central sphere, providing additional binding force proportional to its velocity. As a result, a larger $d_1/D$ ratio may be tolerated in forming the dielectric bonds. Indeed, we have successfully assembled colloidal molecules of $P_1$ and 2.5-μm spheres with dielectric bonds, where $d_1/D = 0.54$.

The increased $d_1/D$ value is associated with greater bond angles. For example, at $f = 2000$ Hz, the bond angle is 90°, indicative of a dominant dipolar repulsion that keeps the metallic lobe away from the central sphere (Fig. 3d, bottom). Besides, the large bond angle can be attributed to the up-shifted nonuniform electric field around the 2.5-μm sphere (Supplementary Figs. 8 and 9), which leads to decreased p-DEP attraction between the metallic lobe and the central sphere. Similar to the previous case, the bond angle can be regulated by frequency, reaching 70°, 53°, and 45° when $f = 1500$, 1000, and 800 Hz, respectively (Fig. 3d, e). It is noteworthy that a wider range of bond angles is realized this way.

Furthermore, if the 4-μm central sphere is assembled with $P_2$ which is of smaller sizes for both the polymeric ($d_1 = 1.17$ μm) and the metallic lobe ($d_2 = 1.46$ μm), the bond angle is fixed at around 30° regardless of the frequency (Fig. 3e). There, both lobes are able to touch the central sphere; for the metallic lobe, the p-DEP attraction by the central sphere overrides the dipolar repulsion even at the high frequency ($f = 2000$ Hz).

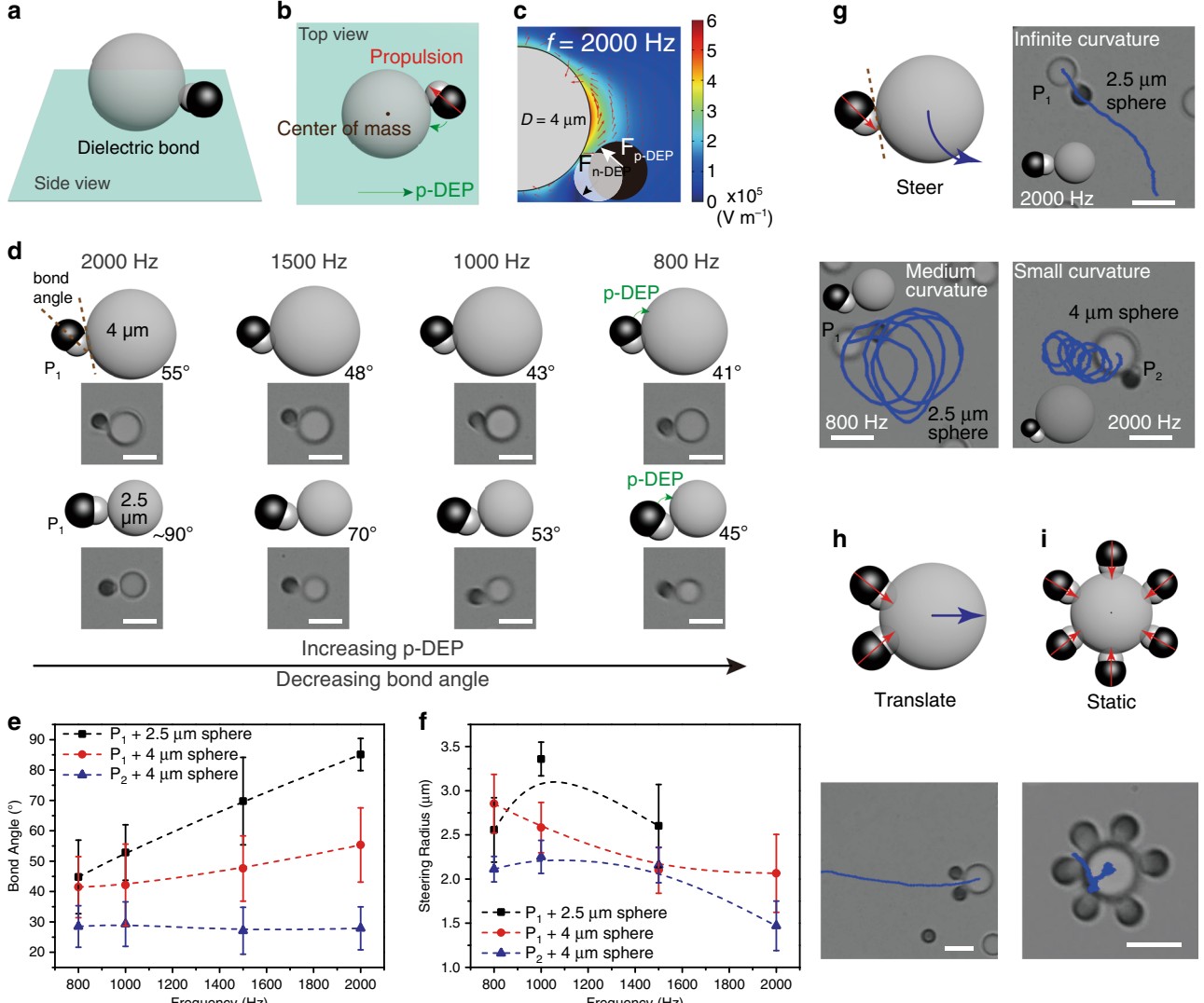

**Fig. 3 Colloidal molecules with dielectric bonds. a** and **b** Cartoon showing the side and top view of an AB colloidal molecule featuring the dielectric bond, formed by a metallodielectric patchy particle $P_1$ and the central dielectric sphere ($D = 4\,\mu m$). The bond angle is defined as the angle between the tangent of where particles touch and long axis of the patchy particle (in $x$–$y$ plane). **c** Finite-element analysis showing that the bonding is favorable due to n-DEP attraction for the dielectric lobe ($x$–$z$ plane), assisted by p-DEP from the metallic lobe. Red arrows represent the direction of electric field lines. Color bar: electric field strength. **d** The bond angle decreases when reducing the AC frequency which increases p-DEP. **e** AC frequency-dependence of bond angles for colloidal molecules assembled by patchy particles $P_1$, $P_2$, and 2.5-/4-μm central spheres as labeled. **f** The radii of steering of the colloidal molecules are determined by their bond angle and the propulsion velocity of the constituent patchy particle, both adjustable by AC frequency (labels are the same as in **e**). **g** Schematic and optical microscope images showing the steering trajectories of colloidal molecules. A wide range of curvature (infinite, medium, and small) are achieved. **h** Colloidal molecule shows translational trajectory when more ligand patchy particles join the assembly (AB$_n$, $n \geq 2$). **i** The saturated colloidal molecules are somewhat static due to balanced forces. Scale bar: 4 μm. The error bars refer to the standard deviation of values for multiple experiments.

As patchy particles within an assembly maintain the propensity to swim (the direction labeled by small red arrow in Fig. 3), the colloidal molecules show intriguing dynamics. When the bond angle is 90°, the assembly follows a pure translational motion (i.e., infinite curvature) as if the patchy particle is pushing the sphere (Fig. 3g). In a more typical case where the bond angle is smaller than 90°, the direction of the patchy particle propulsion does not pass the mass center of the central sphere, therefore leading to steering clusters (Fig. 3b, f, g). The curvature of steering is determined by the bond angle as well as the velocity of the patchy particle propulsion (or equivalently, the propulsion force). They can be regulated by AC frequency and the shapes of the patchy particles (Fig. 3f, g, Supplementary Movie 6 and Discussion 3).

When more patchy particles are in the vicinity, the formation of high-order colloidal molecules (AB$_n$, $n = 2$–6) is observed. For example, two $P_1$ can bind with one 4-μm sphere yielding the AB$_2$ molecules. The patchy particles preferentially slide to one side of the central sphere and together push the sphere to propel. The mode of the propelling motion has switched from steering (for AB) to pure translation (for AB$_2$) (Fig. 3h). The propulsion velocity is decreasing as AC frequency is increasing (see Supplementary Discussion 3). The motion is halted when the periphery of the central sphere is saturated by patchy particles (Fig. 3i). We also note that the high-order molecules are only stable when the dielectric bonds are strong enough to resist the additional dipole repulsion imposed by neighboring patchy

particles. For example, only AB to $AB_3$ are formed by $\mathbf{P_1}$ and the 2.5-μm sphere.

As we have shown, minor adjustment in AC frequency can have a profound impact on the assembly and dynamics of colloidal molecules. Apart from the influence of DEP, we can further understand this by superimposing the involved nonlinear electrokinetic mechanisms, EHD and ICEP, of two or multiple particles within an assembly, on the basis that both EHD and ICEP describe induced charges in diffusive layers that generate electroosmotic flows in response to the applied field[55,62]. For an assembly of colloidal molecule, the total flow $U = U_{ICEP} + U_{EHD}$, where $U_{ICEP} \propto E^2$ and $U_{EHD} \propto E^2 f^{-1}$. As we decrease $f$, $U_{ICEP}$ remains constant, whereas $U_{EHD}$ is increased, for both the dielectric lobe of the patchy particle and the central sphere. Because the flow directions for EHD and ICEP is aligned in clusters with dielectric bonds, the bonding and dynamics are enhanced, and are greater at low frequencies (e.g., propulsion velocity of $AB_n$ is inversely proportional to $f$). A more detailed discussion is available in Supplementary Discussion 3.

**Metallic colloidal bond**. Colloidal molecules featuring metallic bonds are explored next. We hypothesize that the metallic lobe of the patchy particle should be of a proper size to feel a p-DEP attraction that is sufficiently strong to suppress other repulsive forces (e.g., dipolar and EDL). Meanwhile, the dielectric lobe is repelled from (or only slightly attracted by) the central sphere (Fig. 4a, b). As expected, when we use the 4-μm spheres with patchy particle $\mathbf{P_3}$, which features a large dielectric lobe ($d_1 = 1.60$ μm) and a comparably sized metallic lobe ($d_2 = 1.63$ μm), colloidal molecules $AB_n$ ($n = 1–7$) with stable metallic bonds are formed at AC $f \leq 1000$ Hz (Fig. 4c). A good size match with the central sphere is also essential (i.e., the size selectivity). For example, $\mathbf{P_3}$ cannot bind to central spheres that are smaller (e.g., 3.6 μm), where p-DEP is decreased due to the relatively weaker electric field around the 3.6-μm central sphere, together with the stronger induced dipolar repulsion (Supplementary Fig. 9, Movie 7).

Apart from the competition between p-DEP attraction and the dipolar repulsion, a closer look at the metallic bond reveals a delicate balance between the flow-induced forces. On one hand, because individual patchy particle propels by ICEP with its dielectric lobe facing forward, a direction that is opposite to the metallic bond, it tries to escape the assembly which weakens the bond. On the other hand, the EHD flow is always in favor of a bond regardless of the bond nature (Fig. 4a, bottom). With this understanding, we show that the metallic bond can be switched on and off reversibly by a minor change in the field frequency. For example, at $f = 2000$ Hz, no bonds are formed (Supplementary Movie 7); at $f = 800$ Hz, bonds establish as both EHD flow and p-DEP force increase drastically while the magnitude ICEP effect stays roughly the same.

The contrary between the direction of patchy particle propulsion (ICEP) and that of the metallic bond has led to distinct dynamics of the assembly, both during and after the bond formation. As shown in Fig. 4d, the patchy particle on the move approaches and touches the central sphere by its dielectric lobe. We note here that the formation of a dielectric bond is possible, but not energetically as favorable as the metallic bonds in which the patchy particle can be lifted slightly to maximize the p-DEP attraction (Supplementary Fig. 10). As a result, the patchy particle continues to slip along the rim of the central sphere and adjusts its orientation, until a metallic bond is completely formed (Supplementary Movie 7).

In cases where there are already multiple patchy particles attached to the same sphere (e.g., $AB_5$), an incoming patchy particle needs to overcome the repulsion from the existing ligands, lands on the sphere, and finally turns to form a new bond (Fig. 4e, highlighted in red circle). The dipolar repulsion between patchy particles is observed to be strong at low frequency (Supplementary Fig. 11, Movie 8). A patchy particle may miss the colloidal molecules as it is repelled by the existing ligand (Fig. 4e, blue circle). This repulsion can also force some of the bound ligands to disassemble, resembling a bond substitution process in chemical reactions (Supplementary Fig. 12). The dynamic processes for the formation of other selected colloidal molecules are demonstrated in Supplementary Figs. 13 and 14.

To have a clearer picture of the dynamics and particle interactions prior to the bond formation, we track the velocity of patchy particles as they are traveling from a distance and bind with the central sphere. The velocity–distance plots for the formation of the $AB_2$ and $AB_6$ molecules are shown in Fig. 4f. As can be seen, both traces feature a velocity increase when patchy particles are about 1 μm away from the sphere, presumably due to the presence of the inward EHD flow by the sphere that pulls the patchy particle close quickly. For $AB_6$, the repulsion of the bound patchy particles imposed on the incoming particle is revealed by a decrease in its velocity at ~2 μm. Similar patterns can be found for other colloidal molecules (Supplementary Fig. 15).

Unlike colloidal molecules with dielectric bonds, in which the activity of patchy particles is translated to the assembly, with metallic bonds, the whole clusters stay rather static, regardless of the number of patchy particles (Fig. 4g). The ICEP of patchy particles and EHD of the assembled colloidal molecules are balanced ($U = U_{ICEP} + U_{EHD}$, yet ICEP and EHD have opposition flow directions). A control experiment verifies this argument: colloidal molecules with a central sphere and a purely metallic sphere (possessing EHD but not ICEP) can propel with enhanced velocities (Fig. 4h).

**Phase diagram**. As demonstrated, the design principles for colloidal bonds with selectivity and directionality are encoded in the shape and material heterogeneity of the patchy particles and are controlled by the frequency of the electric field. A phase diagram is constructed to summarize our experimental results of assembly as we systematically tune the relevant variables ($f$ and size ratio between the dielectric and central sphere, $d_1/D$, $D = 4$ μm). Regions of no assembly, and those featuring dielectric bond and metallic bond are marked and accompanied by corresponding microscope images (Fig. 5). For example, dielectric bonds occur as long as the dielectric lobe is small enough; metallic bonds are only stable at low frequency when the dielectric lobe is large enough (Supplementary Movie 9). Notably, there is a small region for mixed bonds where both bond types are possible in the same assembly. In this region, the directionality is weakened (Supplementary Movie 9). Although we focus our discussion on $d_1/D$ for simplicity, we supplement that other shape parameters matter too. For example, the particle's aspect ratio will determine the real size and exposed surface of the dielectric and metallic lobes and therefore the static and dynamic interactions. These together establish the final bond properties.

**Integration with biological systems**. We envision that the concept of active colloidal molecules and selective directional bonds can be applied to biological "particles" such as living cells. We expect a two-fold significance for such endeavor. First, synthetic particles can be used to bind cells reversibly and in a predetermined manner. They can deliver substances (such as nutrition and drugs) and functions (local heat, electric field, etc.) for selective cell treatment. Second, biological entities have

**Fig. 4 Colloidal molecules with metallic bonds. a** Cartoon showing the side and top view of an AB colloidal molecule with the metallic bond, by patchy particle $P_3$ and the 4-μm central sphere. **b** Finite-element analysis showing the p-DEP attraction between the metallic lobe and the sphere stabilizes the bond. Red arrows represent the direction of electric field lines. Color bar: electric field strength. **c** Optical microscope images of colloidal molecules with metallic bonds ($AB_1$ to $AB_7$). **d** The formation kinetics of a metallic bond, showing that $P_3$ approaches and touches the big sphere with its dielectric lobe, slips along the rim of central sphere and finally turn to bind the sphere with the metallic lobe. **e** In forming high-order colloidal molecules, $P_3$ overcomes long-range repulsion from bound ligand particles and makes space, before it lands with the dielectric lobe and binds to the metallic lobe. **f** Plot showing the velocity of patchy particles during the formation of $AB_2$ (red square) and $AB_6$ (blue circle) colloidal molecules, revealing the influence of EHD and dipolar repulsion. **g** Colloidal molecules with metallic bonds show weak propulsion ability; **h** Replacing patchy particles with gold spheres enables propulsion of the assembled colloidal molecules. Scale bar: 4 μm.

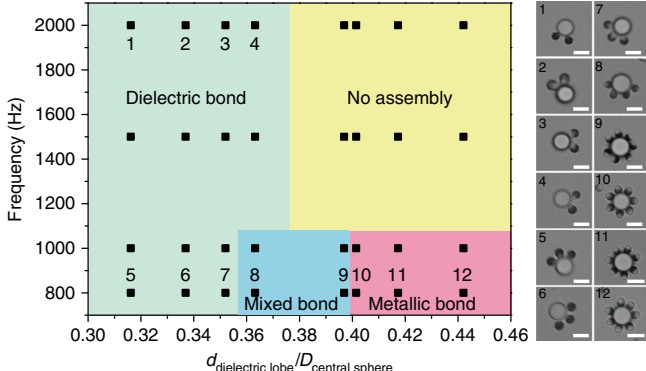

**Fig. 5 Phase diagram of selective and directional colloidal bonds.**
Diagram showing regions of various bonding scenarios, constructed using patchy particles with identically sized metallic lobe but different dielectric lobes and assembled with 4-μm central spheres at various AC frequencies. Scale bar: 4 μm.

various shapes (spheres, rods, irregular shapes, etc.) and dynamics (such as softness) that may be readily available, and they can serve as a template where the self-assembly of synthetic particles can be tuned[63,64].

We first use *E. coli*, rod-shaped bacterial several hundred nanometers in radius and a few micrometers in length, to assemble with dielectric spheres ($D = 4$ μm). When the electric field is on at $f = 300$ Hz, all the bacteria lie on the substrate perpendicular to the field direction (Supplementary Movie 10). The lying bacteria then couple with the central sphere by fitting their tip into the n-DEP trap beneath the sphere. A non-90° bond angle is overserved when multiple bacteria are assembled to the same sphere due to dipolar repulsion. The assembly, integrating both synthetic and biological entities, also propels under the EHD mechanism with the bacteria pushing the sphere (Fig. 6a). We note that this system can serve as effective bacterial transporting system: the sphere can pick up and release the cargo (bacterial) with electric field. Due to the large aspect ratio, the bacteria stand up in parallel to the electric field direction at high frequency, such

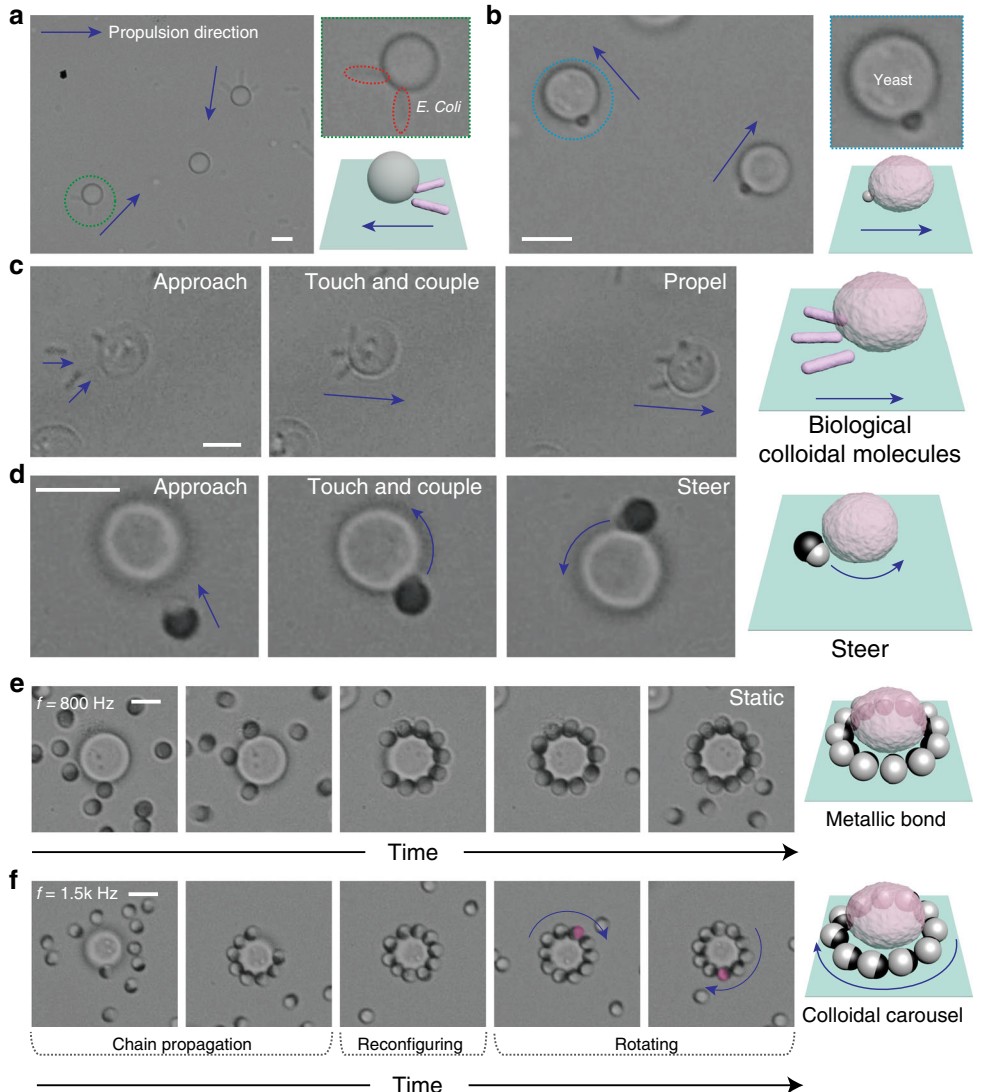

**Fig. 6 Hybrid colloidal molecules by living cells and synthetic particles. a** Self-propelled colloidal molecules assembled by *E. coli* and dielectric sphere ($D = 4 \,\mu$m). **b** Self-propelled colloidal molecules by yeast cells and dielectric sphere ($d = 1 \,\mu$m). **c** Yeast cell and *E. coli* assemble to form purely biological colloidal molecules, which propels under the EHD mechanism. **d** Yeast cell and $P_1$ assemble to form steering colloidal molecules following an approach-touch-steer assembly process. **e** Time lapses showing the formation of colloidal molecules with metallic bond by patchy particle $P_4$ and yeast cell ($f = 800$ Hz). **f** $P_4$ and yeast cells assemble into a colloidal carrousel when propulsion directions of patchy particle synchronize ($f = 1500$ Hz). Scale bar: 4 $\mu$m.

as $f \geq 400$ Hz; strong repulsive interactions prevent any assembly (Supplementary Movie 10).

Next, yeast cells which are roughly spherical in shape and 3–5 $\mu$m in diameter are used to replace the central particle and assemble with various ligands. The structures and dynamics of the resulting assemblies have mostly resembled what have been observed based on purely synthetic particles, but also showed intriguing, surprising phenomena. Specifically, when 1-$\mu$m particles are used, they couple with the yeast forming AB-type molecules that propels with the cell facing forward, characteristic of the EHD mechanism (Fig. 6b, Supplementary Movie 11). This also works for *E. coli*, forming yeast–*E. coli* clusters that propel (Fig. 6c, Supplementary Movie 12). When patchy particle $P_1$ is in use, steering clusters are observed by forming the dielectric bond (Fig. 6d, Supplementary Movie 13). In both cases, the colloidal bonds are relatively weak, as evidenced by the lack of high order molecules, and can be attributed to the decreased n-DEP force due to the differences between living cells and synthetic spheres such as larger

permittivity (permittivity is around 30[65], that of polymer is 2, see Supplementary Discussion 4).

The metallic bonds are realized when patchy particle $P_4$ is employed. At $f = 800$ Hz, the yeast cell catches nearby patchy particles and grows up to high-order colloidal molecules overtime (Fig. 6e, Supplementary Movie 14). Before the surrounding of the cell is saturated, the patchy particles attached are somewhat dynamic, constantly sliding along the cell rim and changing the bond angle (the metallic bonds are kept). This is perhaps due to the irregular shape of the cell that produces some fluctuation in p-DEP around the cell equator. The final assembly features a fully packed layer of patchy particles, which are closely touching each other, static, and highly orientated. In contrast, patchy particles never pack tightly around a synthetic sphere. To account for this, we argue that the cell may deform or squeeze a bit to accommodate more ligands, whereas synthetic spheres cannot. This inspires future exploration in which deformable microgels and capsules can be exploited for controlling particle assemblies using the scheme presented.

When the AC frequency is increased to $f = 1500$ Hz which weakens the p-DEP, the metallic bond angles are smaller than 90°, and in some cases, both the dielectric and metallic lobe are touching the yeast cell. As the cluster develops, all the patchy particles slide to one side of the yeast cell forming a propagating chain. Finally, when the yeast cell is saturated by $\mathbf{P_4}$, the patchy particles synchronize and start to propel around the cell giving rise to a "colloidal carrousel" structure that rotates (Fig. 6f). The formation of such collective motions is a stochastic process and works only if most patchy particles align their directions. Otherwise, patchy particles are stuck leading to a static assembly (Supplementary Movie 15). Since we have not discovered similar dynamics in the assembly of synthetic particles, this suggests that the irregular shape and uneven surface of the yeast cells could play a critical role when the p-DEP force is weak. The bond directionality is also compared (Supplementary Fig. 16), as we change the AC frequencies. Finally, we note that the formation of metallic bond is highly selective toward the size of the cells (Supplementary Fig. 17). In conjunction with the fact that most cells (99.8%) are viable during the course of assembly (i.e., a few minutes) (Supplementary Fig. 18), this serves as the basis for applications such as selective cell treatment, etc.[66].

## Discussion

The development of colloidal science has witnessed two major excitements in the last decade. One is the introduction of patchy colloids with directional interaction which enables complex open structures. The other is the realization of active, self-propelled particles with enhanced motions and controlled trajectories, mimicking those of living organism. Here, we present a merger of the two exciting aspects by demonstrating active assemblies featuring selective and directional bonds. The advantages of our strategy can be considered from two perspectives. On one hand, we are making structures by assembling patchy particles, but the particles are now active and able to propel. The extra energy carried by the patchy particles could facilitate the formation of new structures, or avoid those kinetically trapped intermediates; they may fix defects too[67]. On the other hand, we are engineering micromotors, but instead of a monolithic particle, the motors are assembled in nature which coordinate the dynamics of the constituent particles. This stretches the limit of complexity one can integrate in one motor unit since all the different parts simultaneously come together. In both cases, our method broadens the scope of colloidal structures and dynamics one can possibly achieve.

Centered on selective and directional bonds, we have demonstrated the assembly of discrete colloidal molecules consisting of spheres and patchy particles. They show various structures and dynamics as a result of the bonding scheme introduced. Looking forward, a great challenge is how to obtain active 2D and 3D structures possessing non-close-packing lattices and enhanced dynamics, such as a living crystal with, for example a cubic lattice, or a Kagome lattice by self-propelled particles[68]. In these examples, we note that the combination of selective directional bonds and particle activity is even more critical and requires complex particles to carry all the necessary information. The use of particles with multiple patches and tunable shapes and compositions may provide a solution[59]. In all cases, fine control of the inter-particle interactions, including both the static and dynamic ones, is key.

The interesting findings arisen from the use of bacterial and living cells make our systems promising in bio-medical application and organ-on-a-chip devices. For instance, noble metal nanoparticles are widely utilized in photo-thermal therapy and have proved to be efficient in cancer therapy[69]. Our system provides a selective and straightforward method to bringing noble metal in close contact with the cell. Other synergistic functions may be installed at the dielectric lobe. Besides, the use of biological entities to guide the assembly of synthetic particles in a driven system demands more attention, as unexpected structures and dynamics may emerge, as we have discovered in our systems.

## Methods

**Particle synthesis.** The dielectric spheres of various sizes are synthesized by emulsion polymerization of 3-(trimethoxysilyl)propyl methacrylate (TPM) using ammonia as catalyst and azobisisobutyronitrile (AIBN) as initiator. A comonomer, 3-chloro-2-hydroxypropyl methacrylate (CHPMA) is used to make chlorine-functionalized TPM spheres. The metallodielectric patchy particles are synthesized using a surfactant-aided encapsulation-dewetting method, by which patchy particles of various shapes (size of the dielectric lobe and the metallic lobe) are obtained. Briefly, chlorine-functionalized TPM (Cl-TPM) microspheres are partially encapsulated by plain TPM, with the extent of encapsulation determined by the amount of Triton X-100 added in the system. The Cl-TPM lobe can then be site-specifically coated with a thin layer of gold and serves as the metallic lobe; the plain TPM part is the dielectric lobe.

***E. coli* and yeast cells culture and assembly.** *E. coli* are routinely cultured in Luria-Bertani (LB) medium (Sigma Aldrich, L3522), which consists of 10 g L$^{-1}$ tryptone, 5 g L$^{-1}$ yeast extract, and 5 g L$^{-1}$ NaCl. The culture is shaken at 230 rpm at 37 °C until OD reaches 0.7. Yeast cells are routinely cultured in yeast extract–peptone–dextrose (YPD) medium (Sigma Aldrich, Y1375, 10 g L$^{-1}$ yeast extract, 20 g L$^{-1}$ bacteriological peptone, and 20 g L$^{-1}$ glucose). The resulting *E. coli* and yeast cells are transferred into DI water by centrifugation and redispersion processes (three times) and mixed with synthetic particles. Immediately, a total of 25 μL of the mixed suspension is injected into the chamber made by ITO glass and AC electric field is applied to perform the assembly. The viability of cells after assembly experiment are checked by extracting cells and stained with Trypan blue, which diffuse in dead cells but not live ones. The number of viable and dead cells are counted based on optical microscope images. Cells are subject to electric field for the course of 3, and 30 min for the analysis.

**Self-assembly of colloidal molecules under AC electric field.** Particles of interest (spheres, patchy particles, cells, etc.) are mixed beforehand and are charged in a device chamber sandwiched between two conductive ITO-coated glasses. The ITO glasses are also sputter-coated with a thin layer (5 nm) of iridium to prevent particle sticking. The height of the chamber is about 50 μm. The particles are allowed to settle to the bottom of the device to realize a quasi-two-dimensional (Quasi-2D) experimental system. AC electric fields is then applied between the two ITO glasses using a function generator (RIGOL, DG1022U), with the field direction perpendicular to the ITO glass plane. The peak-to-peak voltage ($V_{pp}$) ranges from 5 to 15 V and the frequency $f$ is from 300 to 4000 Hz. The medium used is DI water. The structures, their formation kinetics and dynamics are directly observed and video-taped on a Nikon Eclipse Ti-2 inverted microscope equipped with a Nikon N7000 DSLR camera. Some of the images were digitally post-processed to improve brightness and contrast.

**Particle tracking.** The propulsion trajectories of particles and assemblies are tracked and processed by Trackpy package with Python.

**Numerical simulation.** COMSOL Multiphysics (Version 5.4a) is employed to calculate the electric field strength distribution around the central dielectric sphere sitting near the bottom electrode. As we have shown, the AC frequency is critical to control the directional bonding, which is rarely discussed in previous model. To find out the effect of frequency on the field distribution, we mainly follow the method proposed by Ristenpart et al.[70]. We consider a thin diffuse layer around the particle surface where the conductivity is much higher than intrinsic material and its influence on the field strength distribution at different frequencies applied. In the model, we define a rectangle, whose size is 200 μm × 100 μm, and place a sphere 4 μm in diameter on the bottom substrate. We use the Laplace equation module and apply the Gauss's Law

$$\nabla^2 \varphi = 0 \tag{1}$$

as the domain condition and the current balance

$$\nabla \varphi \cdot \mathbf{n} = -\lambda \nabla_s^2 \varphi \tag{2}$$

as the particle boundary condition. The applied field strength is set as $5 \times 10^4$ V m$^{-1}$ and top electrode is grounding; $\lambda$ is the dimensionless complex surface conductivity.

## Data availability

Raw data for all figures, codes for particle tracking, and COMSOL simulation are available from the corresponding author upon request.

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

## Acknowledgements

We thank Dr. Ying Li for providing *E. coli* and Gaofei Tian and Prof. David Xiang Li for providing yeast cells as well as their help in materials handling. We thank Prof. Gi-Ra Yi and Dr. Taehee Kang (SKKU, Korea) for their help with the Comsol simulation. Y.F.W. acknowledges support from start-up fund of The University of Hong Kong. This project is also partially supported and the Early Career Scheme (ECS) from the Research Grants Council (RGC) of Hong Kong (Project number: 27303817) and Croucher Innovation Award 2019 (Croucher Foundation, Hong Kong).

## Author contributions

Z.C.W. and Y.F.W. conceived the project. Z.C.W., Z.S.W., and J.H.L. performed the particle synthesis and assembly experiments. Z.S.W. and Z.C.W. performed the Comsol simulation. Z.C.W., Z.S.W., J.H.L., and C.H.T. did the particle tracking and analysis. Z.C.W. and Z.S.W. performed the *E. coli* and yeast assembly experiments. Z.C.W., Z.S.W., and Y.F.W. wrote the manuscript.

## Competing interests

The authors declare no competing interests.
