## [Peer Review File · Nature Communications]

Reviewers' comments:

Reviewer #1 (Remarks to the Author):

This paper describes the assembly and propulsion of colloidal particles in AC electric fields using dielectrophoretic forces and electrohydrodynamic flows. In the simplest realization, two dielectric spheres of sufficiently different radii assemble in the field to produce an asymmetric dimer that moves across the electrode via electrohydrodynamic flows. Collections of smaller satellite particles assemble around a larger sphere to form “molecules” with fluctuating propulsion velocity. When the satellite particles are themselves asymmetric (here, metal-polymer heterodimers), their orientation relative to the large sphere is shown to depend on particle size/geometry and on the frequency of the AC field. This effect provides a basis for different types of “bonds” (dielectric or metallic) that describe the orientation of the satellite particles with respect to the central sphere. The resulting assemblies show different propulsion modalities (linear or circular trajectories) that can be specified by changing the frequency. These dielectrophoretic assemblies can be created using other “particles” including living cells.

This paper does a nice job in exploring how dielectrophoretic assembly and electrokinetic propulsion can be combined to create motile particle assemblies powered by AC electric fields. This work builds on a body of prior work that largely anticipates the present observations: Wu and co-workers have demonstrated how particles can assemble and propel under similar conditions by the same basic mechanisms (refs. 50, 54). The specific geometry of one large sphere surrounded by multiple satellite particles was recently described by Sitti et al. (ref. 51). The new feature of the present work is the use of asymmetric satellite particles, which can “bond” to the larger sphere with different orientations. The ability to specify this bond angle by controlling particle composition/geometry and the ability to modulate this angle (and thereby the propulsion direction) by changing the frequency is arguably the most significant and novel advance of the paper (Figure 3).

I recommend publication in Nature Communications; however, I would appreciate if the Authors could address the following question.

1) What is the difference between induced charge electrophoresis (ICEP) and electrohydrodynamic flow (EHD)? It was my understanding that these two terms describe the same physics --- namely, the standard electrokinetic model (Poisson equation for the potential, Nernst-Planck equation for the ion concentrations, Stokes equation for the fluid velocity). Typically, this model is linearized about the equilibrium double layer (zero applied field). To achieve particle motion and fluid flows from oscillatory driving, one must account for one or more nonlinearities. In ICEP, one considers non-linear contributions to the electric force in the Stokes equations that describe the action of the applied field on that portion of the ionic charge induced by the field (which is non-zero only near interfaces). In this way, the AC field generates steady flows at the particle-fluid and electrode-fluid interfaces, which can result in particle motion. These flows depend (in a complicated fashion) on the frequency of the field, the particle geometry, and the particle composition (e.g., its permittivity at the driving frequency). To first approximation, it may be possible to simply superimpose the field induced disturbances of two or more

particles (i.e., to add the effects of “ICEP” and “EHD”). If this approximate picture was useful in guiding the Authors understanding of the system, then it may be useful also to the reader. Nevertheless, the Authors could provide a clearer discussion of the physical mechanisms that underlie their observations of assembly and propulsion.

Reviewer #2 (Remarks to the Author):

This work presents a new mechanism for the formation of active colloidal molecules. While active colloidal molecules have already been reported (see references below), the key novelty of this work is that the active colloidal molecules are created by using also patchy particles and combining them with biological entities (yeast cells and *E. coli*). The authors present some semi-phenomenological models that explain the observed behaviors and apparently let them predict new behaviors, which they then verify with experiments.

The work is very timely and interesting. I agree with the authors that two of the major trends in soft matter in the last decade are (1) active particles and (2) patchy particles. I think this is an elegant work that combines these two trends. The work is timely and original. I also enjoyed reading the manuscript. I recommend publication once the issues below are addressed.

There are some issues that should be addressed before publication:

1) There has been some previous work on the formation of active colloidal molecules with spherical/non-patchy particles that should be cited. In particular:

Theory:

- R. Soto and R. Golestanian, “Self-assembly of catalytically active colloidal molecules: Tailoring activity through surface chemistry,” *Phys. Rev. Lett.* 112, 068301 (2014)
- A. Ivlev, J. Bartnick, M. Heinen, C.-R. Du, V. Nosenko, and H. Löwen, “Statistical mechanics where Newton’s third law is broken,” *Phys. Rev. X* 5, 011035 (2015)

Experiments:

- F. Schmidt, B. Liebchen, H. Löwen and G. Volpe, “Light-controlled assembly of active colloidal molecules,” *J. Chem. Phys.* 150, 094905 (2019)

2) Why the patchy particles seem to selectively attach only to some yeast cells?

3) Are the bacteria and yeast cells still viable after these experiments?

4) Can there be active colloidal molecules AB with A yeast cells and B bacteria? It'd be very interesting if the the authors try this experiment and report the results.

Very minor point:

5) It's be easier to read the result section if it were divided into subsections.

Reviewer #3 (Remarks to the Author):

The authors assemble defined colloidal clusters consisting of ligand particles surrounding a central particle with high precision using an AC electric field. The AC field induces different electrophoretic effects, which the authors carefully adjust to tailor and control the assembly. Using their high level of understanding of the involved physics, the authors can assemble different number of ligands, can size-selectively assemble and disassemble clusters and control the orientation of patchy particles. They also show how the different arrangements lead to different propulsions of the assembled cluster. As an outlook, they also showcase that biological entities (cells, bacteria) can act as ligand and core particles as well. The manuscript is well written, scientifically sound and showcases a surprising control and versatility in the design of colloidal clusters. As such, it is of interest to a broad scientific community and therefore of interest to Nature Communications. There are, however, several issues that should be addressed before publication.

1) Scope and title

When I read title and abstract, I expected a different direction of the manuscript. Generally, the term bond suggests a more stable attachment, while in the case of the manuscript, the attractive nature disappears in the absence of an electric field. Maybe “dynamic interactions” (or, at least, dynamic bonds) as the authors use in the abstract would be a more fitting term. Second, “directional (bonds)” in the title to me suggest the possibility to define bonds in space. This is not something the proposed methodology is able to provide. The authors use the term in the more literal sense to describe a controlled orientation during attachment – and demonstrate a very good control over this effect. I propose to find a different wording or at least define their definition very early on in the manuscript (ideally in the abstract) and possibly modify the title as well. Finally, in the abstract, the authors introduce “dielectric” and “metallic” bonds. In my mind, this created an expectation of an analogue of a metallic bond in the atomic world, with something similar to delocalised electrons (I was already musing what this may be). Here, I suggest to at least define directly what they mean by the terms to avoid raising wrong expectations.

2) Control over aggregates and statistical evaluation of the data

On page 4, the authors mention that in mixtures of 4 and 1 μ m particles, heterogeneous dimer clusters (AB) form exclusively. Further on (P.8), they state that high-order colloidal molecules (AB_n) are realized

when more ligand particles are available. Did I miss a difference in the experimental conditions between Figure 1 (AB-cluster) and Figure 2 (AB_n) clusters? If not, I find the statement of exclusiveness misleading. More generally, the manuscript focuses a lot on the investigations of individual clusters, which, of course, is necessary for the detailed analysis of bonding and movement. However, I believe that especially in the context of Figs 1 and 2, some statistical data should be provided to answer questions related to the cluster formation efficiency: How often do dimers form if the right stoichiometry is chosen? Will there be exclusively dimers or will the number of ligands be distributed? How will the situation shift with increasing numbers of ligand particles? Providing such numbers is important to avoid appearing to only “cherry-pick” clusters that fit the model.

3) Figure 3

In Figure 3, the authors provide a detailed model on the control of bond angle between ligand and center by the applied frequency (and the size ratio). It would add to the manuscript if real microscopy images would be provided that show that the model proposed in the cartoons match the real system.

4) Miscellaneous

P.4: The authors advertise the use of biological entities with their special features that are not accessible by synthetic systems. They use the example of deformability, which I believe is poorly chosen since this is a property that can be well controlled synthetically (microgels, capsules,...).

P.5. The sentence: “most crucial to such size selective bonding is dielectrophoretic interaction, among others” seems odd (either most crucial, or, interaction among others)

P.6: “avoid the beneath of the central particles” – do the authors mean “avoid being attracted at the bottom part of the central particle”?

P.6 (bottom): “the asymmetrical shape [...] causes a flow, which propels the clusters towards the bigger central particle”. Either the ligand is propelled towards the central particle, or the entire cluster is propelled, but not to the central particle.

Sup. Figure 5: It would be useful to assign the color to the material in the figure caption. It is also a bit confusing that the cartoon uses dark patches for the metal part, while in the SEM, this part is the light one. A description may avoid confusion.

Nicolas Vogel

Referee #1:

This paper describes the assembly and propulsion of colloidal particles in AC electric fields using dielectrophoretic forces and electrohydrodynamic flows. In the simplest realization, two dielectric spheres of sufficiently different radii assemble in the field to produce an asymmetric dimer that moves across the electrode via electrohydrodynamic flows. Collections of smaller satellite particles assemble around a larger sphere to form “molecules” with fluctuating propulsion velocity. When the satellite particles are themselves asymmetric (here, metal-polymer heterodimers), their orientation relative to the large sphere is shown to depend on particle size/geometry and on the frequency of the AC field. This effect provides a basis for different types of “bonds” (dielectric or metallic) that describe the orientation of the satellite particles with respect to the central sphere. The resulting assemblies show different propulsion modalities (linear or circular trajectories) that can be specified by changing the frequency. These dielectrophoretic assemblies can be created using other “particles” including living cells.

This paper does a nice job in exploring how dielectrophoretic assembly and electrokinetic propulsion can be combined to create motile particle assemblies powered by AC electric fields. This work builds on a body of prior work that largely anticipates the present observations: Wu and co-workers have demonstrated how particles can assemble and propel under similar

conditions by the same basic mechanisms (refs. 50, 54). The specific geometry of one large sphere surrounded by multiple satellite particles was recently described by Sitti et al. (ref. 51). The new feature of the present work is the use of asymmetric satellite particles, which can “bond” to the larger sphere with different orientations. The ability to specify this bond angle by controlling particle composition/geometry and the ability to modulate this angle (and thereby the propulsion direction) by changing the frequency is arguably the most significant and novel advance of the paper (Figure 3)

Authors:

We thank the referee for the careful examination of our paper and the positive report. We are glad that the referee finds our work of “*significant and novel advance*”.

Referee #1:

I recommend publication in Nature Communications; however, I would appreciate if the Authors could address the following question.

Authors:

We thank the referee for the recommendation. We have addressed the question raised by the referee, as detailed below.

Referee #1:

What is the difference between induced charge electrophoresis (ICEP) and electrohydrodynamic flow (EHD)? It was my understanding that these two terms describe the same physics --- namely, the standard electrokinetic model (Poisson equation for the potential, Nernst-Planck equation for the ion concentrations, Stokes equation for the fluid velocity). Typically, this model is linearized about the equilibrium double layer (zero applied field). To achieve particle motion and fluid flows from oscillatory driving, one must account for one or more nonlinearities. In ICEP, one considers non-linear contributions to the electric force in the Stokes equations that describe the action of the applied field on that portion of the ionic charge induced by the field (which is non-zero only near interfaces). In this way, the AC field generates steady flows at the particle-fluid and electrode-fluid interfaces, which can result in particle motion. These flows depend (in a complicated fashion) on the frequency of the field, the particle geometry, and the particle composition (e.g., its permittivity at the driving frequency).

Authors:

We thank the referee for the valuable analysis. We agree that ICEP and EDH describe the same physics namely the standard/nonlinear electrokinetic models. They both consider the effect of electric field on ionic charges induced by the field at fluid interfaces. In both cases, the generated flow U scales nonlinearly with the electric field, $U \propto E^2$. A quote from a paper of Wu and co-workers is “The EHD flow is essentially an induced-charge electroosmosis along the electrode” (N. Wu et al., *ACS Appl. Mater. Interfaces* 2014), supporting this point.

Despite the same physical origin, ICEP and EHD are generally considered as two electrokinetic mechanisms, due to that the flow is generated at different interfaces, i.e., the particle-fluid interface for ICEP and the electrode-fluid interface for EHD, and “*depend on the frequency of the field, the particle geometry, and the particle composition*”, as the referee points out. When these details (composition, frequency, etc.) are plugged in, there are some subtle differences of the two mechanisms on a particle, which have been thoroughly analyzed by Wu and co-workers (N. Wu et al., *ACS Appl. Mater. Interfaces* 2014). For example, ICEP for dielectric particles are negligible, whereas EHD has impacts on both metallic and dielectric particles as long as the particle is close to the conducting substrate. Also, in the frequency (f) range studied, ICEP is nearly constant, whereas EHD is approximately proportional to f^{-1} .

Referee #1:

To first approximation, it may be possible to simply superimpose the field induced disturbances of two or more particles (i.e., to add the effects of “ICEP” and “EHD”). If this approximate picture was useful in guiding the Authors understanding of the system, then it may be useful also to the reader. Nevertheless, the Authors could provide a clearer discussion of the physical mechanisms that underlie their observations of assembly and propulsion.

Authors:

The referee has a great point. It is indeed possible to superimpose the different electrokinetic mechanisms of complex particle systems. In fact, Yossifon, Velev and co-workers have recently analyzed the total angular velocity (θ) of a colloidal spinner by the superposition of contribution from multiple propulsion modes including ICEP, EHD and sDEP, i.e., $\theta_{total} = \theta_{ICEP} + \theta_{EHD} + \theta_{sDEP}$ (G. Yossifon, O. D. Velev et al., *Adv. Funct. Mater.* 2018). Also, Wu and co-workers have earlier combined the effect of ICEP and EHD to explain the propulsion of metallodielectric dimers and their frequency dependence (N. Wu et al., *ACS Appl. Mater. Interfaces* 2014).

In response to the reviewer’s suggestion, we have added sentences where we first introduce EHD and ICEP so the readers can be refreshed about the physical mechanisms as well as how they (i.e., the flow) scale with electric field strength and frequency (Page 6, 8, 10). More importantly, we have added a paragraph at the end of the “Colloidal bond directionality” section to discuss the combined effect of EHD and ICEP in determining the behavior of colloidal clusters presented. For example, for colloidal clusters with the “dielectric bond”, the flow direction of EHD and ICEP is aligned, so the propulsion is enhanced and increased as f decreases. For colloidal molecules with the “metallic bond”, the direction of EHD and ICEP is opposite. The result of the competition/cancellation is that the clusters are very static (Page 12-13). In addition, we have analyzed in detail in Supplementary Information how EHD and ICEP is together influencing the dynamics of active patchy particles and the assemblies (Supplementary Discussion and Figure S20, Page S21-24).

Again, we thank the reviewer for this great comment/suggestion, which has directed us to provide a clearer presentation of our system.

Referee #2:

This work presents a new mechanism for the formation of active colloidal molecules. While active colloidal molecules have already been reported (see references below), the key novelty of this work is that the active colloidal molecules are created by using also patchy particles and combining them with biological entities (yeast cells and E. coli). The authors present some semi-phenomenological models that explain the observed behaviors and apparently let them predict new behaviors, which they then verify with experiments.

The work is very timely and interesting. I agree with the authors that two of the major trends in soft matter in the last decade are (1) active particles and (2) patchy particles. I think this is an elegant work that combines these two trends. The work is timely and original. I also enjoyed reading the manuscript. I recommend publication once the issues below are addressed.

Authors:

We thank the referee for the very positive comments, describing our work as “*is timely and interesting*”, “*elegant*” and “*original*”. We are very delighted that the referee has enjoyed reading our paper and has recommended for publication. We are also grateful for the comments/issues that have been raised, which we find inspiring and have addressed below.

Referee #2:

There are some issues that should be addressed before publication:

1) There has been some previous work on the formation of active colloidal molecules with spherical/non-patchy particles that should be cited. In particular:

Theory:

- R. Soto and R. Golestanian, “Self-assembly of catalytically active colloidal molecules: Tailoring activity through surface chemistry,” Phys. Rev. Lett. 112, 068301 (2014).

- A. Ivlev, J. Bartnick, M. Heinen, C.-R. Du, V. Nosenko, and H. Löwen, “Statistical mechanics where Newton’s third law is broken,” Phys. Rev. X 5, 011035 (2015).

Experiments:

- F. Schmidt, B. Liebchen, H. Löwen and G. Volpe, “Light-controlled assembly of active colloidal molecules,” J. Chem. Phys. 150, 094905 (2019).

Authors:

We thank the referee for the suggestion. We have included all the references, Ref. 56, 57, and 58.

Referee #2:

2) Why the patchy particles seem to selectively attach only to some yeast cells?

Authors:

The binding of particles under the presented mechanism shows high selectivity according to their sizes. This is mainly due to the fact that dielectrophoretic and dipole interactions depend on the particle geometries (elaborated in the “Colloidal bond selectivity” section). In the case of cells, the size distribution is wide, so that a portion of cells won’t be able to bind with particles due to size mismatch. Compared to synthetic particles, the selectivity is even more rigorous because the bond between particle and cell is normally weak due to various reasons such as large permittivity of cells (see Supplementary Discussion). This reduces the chance of assembly. Moreover, cells have irregular shape, so patchy particle cannot bind from all directions which further lowers the possibility of forming a bond.

Referee #2:

3) Are the bacteria and yeast cells still viable after these experiments?

Authors:

We thank the referee for this question. We have conducted the viability experiments for Yeast cells and included the data in Supplementary Information (Supplementary Figure S18). Briefly, the assembly condition, i.e., AC electric field at relative low frequencies and strengths has minimum damages to cells in time scales that allow for particle assembly (less than 1 min). For example, upon exposure under AC field of various parameters for 3 min, the viable cells are about 99%. We have added a sentence in the main text to mention this conclusion (Page 20). Longer exposure has led to increased cell death; the viability for 30 min dropped to 89.5%. In fact, cell viability (including experiments for *E. Coli*) under AC electric field have been well studied for other purposes (e.g., separation). We have also cited those references (e.g., Ref. 66).

Referee #2:

4) Can there be active colloidal molecules AB with A yeast cells and B bacteria? It'd be very interesting if the authors try this experiment and report the results.

We thank the referee for this interesting question (it was strange we never thought about this!). We have tried the experiment and Yeast and *E. Coli* indeed form AB-type colloidal molecules that propel under EHD mechanism. We have included the results in Figure 6c, Page 19, and Supplementary Video S11.

Referee #2:

Very minor point:

5) It's be easier to read the result section if it were divided into subsections.

Authors:

We have included subsections for the ease of reading. We thank the referee for the suggestion.

Referee #3:

The authors assemble defined colloidal clusters consisting of ligand particles surrounding a central particle with high precision using an AC electric field. The AC field induces different electrophoretic effects, which the authors carefully adjust to tailor and control the assembly. Using their high level of understanding of the involved physics, the authors can assemble different number of ligands, can size-selectively assemble and disassemble clusters and control the orientation of patchy particles. They also show how the different arrangements lead to different propulsions of the assembled cluster. As an outlook, they also showcase that biological entities (cells, bacteria) can act as ligand and core particles as well. The manuscript is well written, scientifically sound and showcases a surprising control and versatility in the design of colloidal clusters. As such, it is of interest to a broad scientific community and therefore of interest to Nature Communications. There are, however, several issues that should be addressed before publication.

Authors:

We thank the referee for carefully examining our work and the positive comments. We are very glad that our paper is considered “*well-written*”, “*scientifically sound*”, and “*of interest to a broad scientific community and therefore of interest to Nature Communications*”. We also appreciate that the referee has raised important questions as well as valuable thoughtful suggestions that have directed us to revise the paper for improved readability and quality.

Referee #3:

1) Scope and title.

When I read title and abstract, I expected a different direction of the manuscript. Generally, the term bond suggests a more stable attachment, while in the case of the manuscript, the attractive nature disappears in the absence of an electric field. Maybe “dynamic interactions” (or, at least, dynamic bonds) as the authors use in the abstract would be a more fitting term.

Authors:

We thank the referee for the comment. We can understand the referee’s concern, but wish to humbly explain our rationale for deciding the wording of the title.

First, “colloidal molecules” have been well acknowledged in the field to describe small clusters of particles (E. Duguet, et al., *Chem. Soc. Rev.* 2011; D.J. Pine, et al., *Nature* 2012; C. Fan, *Nat. Mat.* 2019.), despite that they are still very different from real molecules and that “cluster” may already be the proper term for what they really are. Accordingly, “colloidal bond” has been used to describe the bridging of particles within a “colloidal molecule”, analogous to the “C-H bond” in a “methane molecule”. In our case, we wish to use the “colloidal bond” to describe the action of assembly the

“colloidal molecules”, as a result of many specific interactions. Here, many dynamic interactions contribute to the bonding between particles.

Second, in the context of “active colloids”, the dynamic behavior of particles is of interest, which is out-of-equilibrium due to constant energy consumption, that is, under external stimuli (here energy is provided by the electric field). We argue active system is slightly different from the traditional “stimuli-responsive materials”, which may focus more on property switching when the stimulus is turned on and off. Certainly, in our system, nothing interest will be observed when AC field is off. Therefore, we believe that “bond” could still be an appropriate term to use, as the hypothesis is that the electric field is on all the time. Following the referee’s suggestion, we have added “...dynamic colloidal bonds...” in the abstract and the main text where it is suitable. As for the title, we humbly suggest not to use “dynamic bond” just to keep the title concise; the “dynamic” nature of the system is to some extent covered by the word “active” in the title.

Finally, the key advance of our work is the realization of “selective and directional bond”, which we intentionally include in the tile to relate it to “specific directional bonding”, a key concept that was previously enabled by us and others by employing patchy particles (e.g., D.J. Pine, et al., *Nature* 2012;). Earlier, it was focused on equilibrium assembly systems, while in this work, it is focused on active systems. (This also partially responses to the referee’s next comment.)

We thank the referee again for raising this issue. We humbly wish the referee can understand our thoughts.

Referee #3:

Second, “directional (bonds)” in the title to me suggest the possibility to define bonds in space. This is not something the proposed methodology is able to provide. The authors use the term in the more literal sense to describe a controlled orientation during attachment – and demonstrate a very good control over this effect. I propose to find a different wording or at least define their definition very early on in the manuscript (ideally in the abstract) and possibly modify the title as well.

Authors:

We thank the reviewer for the suggestion/comment. We have partially responded to it above. We agree that some terms are not vigorously aligned with those defined in chemistry but are somewhat accepted in the field. To avoid possible wrong expectations, we have followed the referee’s suggestion to define/explain those terms early in the paper. We have also added more sentences to elaborate on those terms, and to note the differences. Please see Page 1, 3, and 9.

Referee #3:

Finally, in the abstract, the authors introduce “dielectric” and “metallic” bonds. In my mind, this created an expectation of an analogue of a metallic bond in the atomic world, with something similar to delocalised electrons (I was already musing what this may be). Here, I suggest to at least define directly what they mean by the terms to avoid raising wrong expectations

Authors:

We thank the referee for this important reminder. We have modified them to “dielectric colloidal bond” and “metallic colloidal bond” when they first appear. We have also added a sentence to help with the definition in the context. Please see Page 1 and 9.

Referee #3:

2) *Control over aggregates and statistical evaluation of the data.*

On page 4, the authors mention that in mixtures of 4 and 1 μ m particles, heterogeneous dimer clusters (AB) form exclusively. Further on (P.8), they state that high-order colloidal molecules (AB_n) are realized when more ligand particles are available. Did I miss a difference in the experimental conditions between Figure 1 (AB-cluster) and Figure 2 (AB_n) clusters? If not, I find the statement of exclusiveness misleading. More generally, the manuscript focuses a lot on the investigations of individual clusters, which, of course, is necessary for the detailed analysis of bonding and movement. However, I believe that especially in the context of Figs 1 and 2, some statistical data should be provided to answer questions related to the cluster formation efficiency: How often do dimers form if the right stoichiometry is chosen? Will there be exclusively dimers or will the number of ligands be distributed? How will the situation shift with increasing numbers of ligand particles? Providing such numbers is important to avoid appearing to only “cherry-pick” clusters that fit the model.

Authors:

We thank the referee for this point. By using “exclusively” here we wish to express that only A and B will bind to form a cluster, and there will be no AA or BB clusters. To control the number of ligand B per central particle A in the assembly, i.e., the value of “*n*”, we mix A and B with different B/A feed ratio before the assembly.

To fix the issue, we have added sentences to elaborate our point (Page 4 and 8). Also, in response to the referee’s request, we have included the statistics in Figure 2e as well as Supplementary Figure S4 to show the number distribution of AB_n clusters as a function of *n*, at various initial B/A ratios. Briefly, AB₁ clusters (dimers) are popular when the B/A ratio is low (such as 0.3); higher-order cluster are observed when B/A ratio is high (such as 13), as illustrated by the revised figures.

Referee #3:

3) *Figure 3. In Figure 3, the authors provide a detailed model on the control of bond angle between ligand and center by the applied frequency (and the size ratio). It would add to the manuscript if real microscopy images would be provided that show that the model proposed in the cartoons match the real system.*

Authors:

We thank the referee for this helpful comment. In response, we have added microscope images of the particle assemblies. Please see the revised Figure 3d.

Referee #3:

4) Miscellaneous

P.4: The authors advertise the use of biological entities with their special features that are not accessible by synthetic systems. They use the example of deformability, which I believe is poorly chosen since this is a property that can be well controlled synthetically (microgels, capsules,...).

Authors:

We thank the referee for raising this valuable point, which has inspired us for the possible future exploration of microgels and capsules for similar experiments, where the sizes and permittivity of particles can be adjusted by an orthogonal stimulus such as heat.

We have revised our statement accordingly. Now it reads “...because biological entities have diverse shapes (such as irregular spheres and rods) and properties (e.g., they are deformable) readily available...” in Page 4, and “This can inspire future exploration in which deformable microgels and capsules are exploited for controlling particle assemblies using the scheme presented” in Page 19.

Referee #3:

P.5. The sentence: “most crucial to such size selective bonding is dielectrophoretic interaction, among others” seems odd (either most crucial, or, interaction among others)

Authors:

We have revised the sentence following the referee’s suggestion. Now it reads “Most crucial to such size selective bonding is dielectrophoretic interaction”.

Referee #3:

P.6: “avoid the beneath of the central particles” – do the authors mean “avoid being attracted at the bottom part of the central particle”?

Authors:

We thank the referee for catching this confusing sentence. After revision it reads “...avoid being attracted at the bottom part of the central particle...”.

Referee #3:

P.6 (bottom): “the asymmetrical shape [...] causes a flow, which propels the clusters towards the bigger central particle”. Either the ligand is propelled towards the central particle, or the entire cluster is propelled, but not to the central particle.

Authors:

We thank the referee for the suggestion. We have fixed the sentence. It now reads “...which propels the clusters with the bigger central particle orienting forward”.

Referee #3:

Sup. Figure 5: It would be useful to assign the color to the material in the figure caption. It is also a bit confusing that the cartoon uses dark patches for the metal part, while in the SEM, this part is the light one. A description may avoid confusion.

Authors:

We really appreciate that the referee has carefully examined our supplementary document. The color of the cartoon is assigned according to optical microscope images of patchy particles presented in the main text. To avoid confusion in the SEM of Supplementary Figure S5 (it's now Figure S6), we have added arrows to show the metallic and dielectric lobe of the particle, as well as a description in the figure caption.

REVIEWERS' COMMENTS:

Reviewer #1 (Remarks to the Author):

The Authors have adequately addressed my previous comment and have added clarifying discussions to the main text to distinguish EHD and ICEO flows.

Reviewer #2 (Remarks to the Author):

I think the authors have addressed the issues raised by the referees satisfactorily. I think the article can now be published in Nat Commun

Reviewer #3 (Remarks to the Author):

The authors have meticulously addressed all remarks and I believe that this paper now stands as a valuable contribution to the fields of active soft matter, self-assembly and colloid and interface sciences. I therefore strongly recommend publication in Nature Communications.

I now understand the discussion and rationale behind the choice of the title and the small modifications in the text are well chosen to clarify the proposed concepts and novelty. I also appreciate the additions to Figs 2 and 3 about statistics and the microscopy data, which really strikingly demonstrate the degree of control the authors exert on their system. Finally, I was intrigued by the all-biological colloidal molecules the authors were able to prepare from the suggestion of reviewer 2, which showcases the potential and versatility of the approach.

Nicolas Vogel